# Glutathione binding to the plant *At*Atm3 transporter and implications for the conformational coupling of ABC transporters

**Chengcheng Fan†, Douglas C Rees\***

Division of Chemistry and Chemical Engineering, Howard Hughes Medical Institute, California Institute of Technology, Pasadena, United States

**Abstract** The ATP binding cassette (ABC) transporter of mitochondria (Atm) from *Arabidopsis thaliana* (*At*Atm3) has been implicated in the maturation of cytosolic iron-sulfur proteins and heavy metal detoxification, plausibly by exporting glutathione derivatives. Using single-particle cryo-electron microscopy, we have determined four structures of *At*Atm3 in three different conformational states: two inward-facing conformations (with and without bound oxidized glutathione [GSSG]), together with closed and outward-facing states stabilized by MgADP-VO$_4$. These structures not only provide a structural framework for defining the alternating access transport cycle, but also reveal the paucity of cysteine residues in the glutathione binding site that could potentially form inhibitory mixed disulfides with GSSG. Despite extensive efforts, we were unable to prepare the ternary complex of *At*Atm3 containing both GSSG and MgATP. A survey of structurally characterized type IV ABC transporters that includes *At*Atm3 establishes that while nucleotides are found associated with all conformational states, they are effectively required to stabilize occluded, closed, and outward-facing conformations. In contrast, transport substrates have only been observed associated with inward-facing conformations. The absence of structures with dimerized nucleotide binding domains containing both nucleotide and transport substrate suggests that this form of the ternary complex exists only transiently during the transport cycle.

**\*For correspondence:**
dcrees@caltech.edu

**Present address:** †Division of Biology and Biological Engineering, California Institute of Technology, Pasadena, United States

**Competing interest:** The authors declare that no competing interests exist.

## Editor's evaluation

Mitochondrial glutathione is an important line of defence against free radical production. The ATP binding cassette (ABC) transporter Atm3 exports oxidized glutathione out of the mitochondria to help maintain a suitable reducing environment. In this study, the authors have biochemically characterized Atm3 and determined four cryo-EM structures exhibiting three different conformational states, revealing new insights into the transport mechanism. This well-executed study will be of broad interest to the membrane biology and transport communities.

## Introduction

The ATP binding cassette (ABC) transporter of mitochondria (Atm) family plays a vital (*Leighton and Schatz, 1995*), but enigmatic, role broadly related to transition metal homeostasis in eukaryotes (*Lill et al., 2014*). The best characterized member is *Saccharomyces cerevisiae* Atm1 (*Sc*Atm1) present in the inner membrane of mitochondria (*Leighton and Schatz, 1995*) and required for the formation of cytosolic iron-sulfur cluster containing proteins (*Kispal et al., 1999*). Defects in *Sc*Atm1 lead to an overaccumulation of iron in the mitochondria (*Kispal et al., 1997*). Atm1 is proposed to transport a

sulfur containing intermediate (*Kispal et al., 1999*) that may also include iron (*Pandey et al., 2019*). It is also likely to transport a similar sulfur containing species from the mitochondria that is required for the cytoplasmic thiolation of tRNA (*Pandey et al., 2018*). While the precise substrate that is transported remains unknown, derivatives of glutathione have been implicated based on their ability to stimulate the ATPase activity of Atm1 (*Kuhnke et al., 2006*).

Structures for Atm family members are currently available for *Sc*Atm1 (*Srinivasan et al., 2014*), the bacterial homolog *Na*Atm1 from *Novosphingobium aromaticivorans* (*Lee et al., 2014*) and human ABCB6 (*Wang et al., 2020*); the pairwise sequence identities between these homologous transporters range from 40% to 46%. These proteins occur as homodimers of half-transporters, where each half-transporter contains a transmembrane domain (TMD) followed by the canonical nucleotide binding domain (NBD) that defines the ABC transporter family. Each TMD consists of six transmembrane helices (TMs) that exhibit the exporter type I fold first observed for Sav1866 (*Dawson and Locher, 2006*); a recent re-classification now identifies this group as type IV ABC transporters (*Thomas et al., 2020*). The translocation of substrates across the membrane proceeds through an alternating access mechanism involving the ATP-dependent interconversion between inward- and outward-facing conformational states. Among the Atm1 family, these conformations have been most extensively characterized for *Na*Atm1 and include the occluded and closed states that provide a structural framework for the unidirectional transport cycle (*Fan et al., 2020*). Structures of *Sc*Atm1 with reduced glutathione (GSH) (*Srinivasan et al., 2014*), and of *Na*Atm1 complexed with reduced (GSH), oxidized (GSSG), and metallated (GS-Hg-SG) (*Lee et al., 2014*), have defined the general substrate binding site in the TMD for the transport substrates.

Plants have been found to have large numbers of transporters (*Hwang et al., 2016*), including *Arabidopsis* with three Atm orthologues, *At*Atm1, *At*Atm2, and *At*Atm3 (*Chen et al., 2007*). Of these, *At*Atm3 (also known as ABCB25) rescues the *Sc*Atm1 phenotype (*Chen et al., 2007*), and has been shown to be associated with maturation of cytosolic iron-sulfur proteins (*Kushnir et al., 2001*), confer resistance to heavy metals such as cadmium and lead (*Kim et al., 2006*), and participate in the formation of molybdenum-cofactor containing enzymes (*Bernard et al., 2009*; *Teschner et al., 2010*). Unlike yeast, defects in *At*Atm3 are not associated with iron accumulation in mitochondria (*Bernard et al., 2009*). While the physiological substrate is unknown, *At*Atm3 has been shown to transport GSSG and glutathione polysulfide, with the persulfidated species perhaps relevant to cytosolic iron-sulfur cluster assembly (*Schaedler et al., 2014*). The ability of *At*Atm3 to export GSSG has been implicated in helping stabilize against excessive glutathione oxidation in the mitochondria and thereby serving to maintain a suitable reduction potential (*Marty et al., 2019*).

To help address the functional role(s) of Atm transporters, we have determined structures of *At*Atm3 in multiple conformational states by single-particle cryo-electron microscopy (cryoEM). These structures not only provide a structural framework for defining the alternating access transport cycle, but they also illuminate an unappreciated feature of the glutathione binding site, namely the paucity of cysteine residues that could potentially form inhibitory mixed disulfides during the transport cycle. A survey of structurally characterized members of the type IV family of ABC transporters, including the Atm1 family, establishes that nucleotides are effectively required for the stabilization of the occluded, closed, and outward-facing conformations. In contrast to the nucleotide states, transport substrates and related inhibitors have only been observed associated with inward-facing conformational states. The absence of structures with dimerized NBDs containing both nucleotide and transport substrate suggests that this form of the ternary complex exists only transiently during the transport cycle.

## Results

*At*Atm3 contains an N-terminal mitochondrial targeting sequence that directs the translated protein to the mitochondria, where it is cleaved following delivery to the inner membrane. Since this targeting sequence consists of ~80 residues and is anticipated to be poorly ordered, we generated three different N-terminal truncation mutants of *At*Atm3 through deletion of 60, 70, or 80 residues to identify the best-behaved construct. Together with the wild-type construct, these three variants were heterologously overexpressed in *Escherichia coli*. The construct with the 80 amino acids deletion showed the highest expression level and proportionally less aggregation by size exclusion chromatography (*Figure 1—figure supplement 1*) and hence was used for further functional and structural studies.

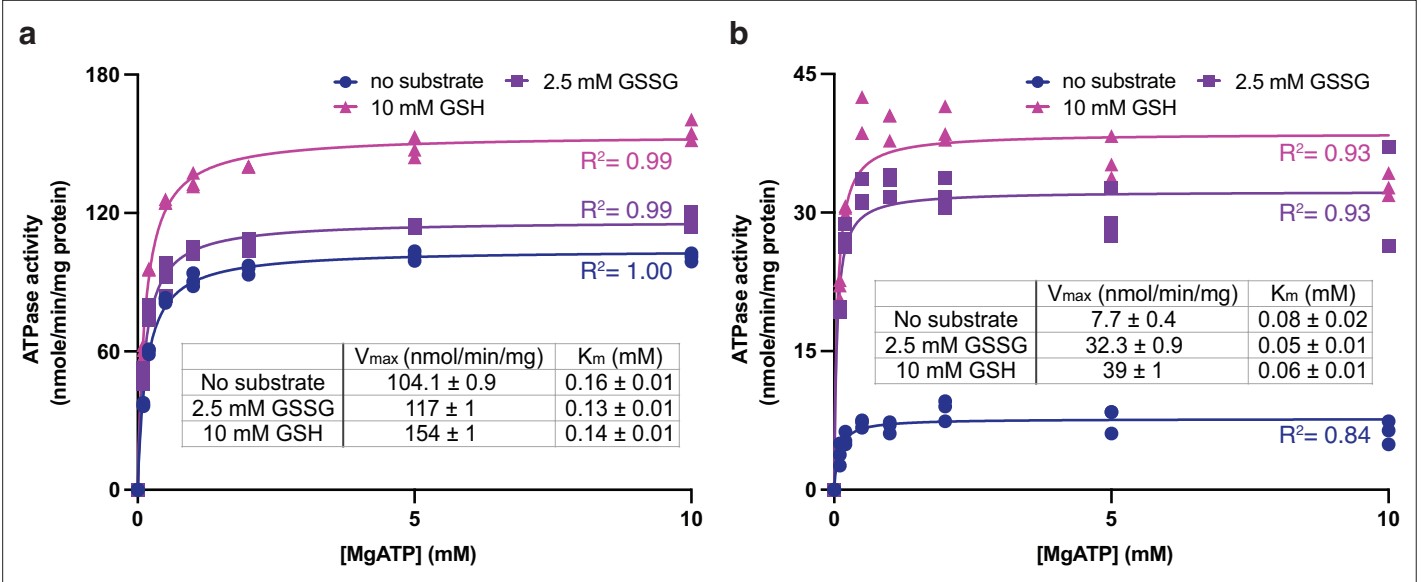

**Figure 1.** ATPase activities of *At*Atm3. ATPase activities measured in (**a**) the detergent *n*-dodecyl-β-D-maltoside (DDM) and (**b**) nanodiscs formed by membrane scaffolding proteins (MSP) and the lipid 1-palmitoyl-2-oleoyl-glycero-3-phosphocholine (POPC). The ATPase activities were measured in the absence of substrate (●), at 2.5 mM GSSG (■) and 10 mM GSH (▲). The corresponding values of $V_{max}$ and $K_m$ in different substrate conditions derived from fitting to the Michaelis-Menten equations are indicated. Each condition was measured three times with the individual data points displayed.

The online version of this article includes the following source data and figure supplement(s) for figure 1:

**Source data 1.** Numerical data for the graphs depicted in *Figure 1a and b*.

**Figure supplement 1.** *At*Atm3 constructs.

## ATPase activities

Using the 80-residue truncation construct, *At*Atm3 was purified in the detergent *n*-dodecyl-β-D-maltoside (DDM) and reconstituted into nanodiscs formed from the membrane scaffolding protein (MSP) 1D1 and the lipid 1-palmitoyl-2-oleoyl-glycero-3-phosphocholine (POPC). The ATPase activity of this construct was measured as a function of MgATP concentration in the absence and presence of either 2.5 mM GSSG or 10 mM GSH, which approximate their physiological concentrations in *E. coli* (*Bennett et al., 2009*). The rate of ATP hydrolysis was determined by measuring phosphate release using a molybdate-based colorimetric ATPase activity assay (*Chifflet et al., 1988*). The basal ATPase activity, measured in the absence of glutathione derivatives, was significantly higher in detergent than in nanodiscs (104 vs. 7.7 nmol/min/mg, respectively; *Figure 1ab*), while the apparent $K_m$s for MgATP were within a factor of two (~0.16 and 0.08 mM, respectively). The ATPase activity of *At*Atm3 is stimulated by both 2.5 mM GSSG and 10 mM GSH, but the extent of stimulation depends strongly on the solubilization conditions. In nanodiscs, the ATPase rates increase to 32 and 39 nmol/min/mg with 2.5 mM GSSG and 10 mM GSH, respectively, for an overall increase of 4–5× above the basal rate. The ATPase rates for *At*Atm3 in DDM also increase with GSSG and GSH, to 117 and 154 nmol/min/mg, respectively. Because of the higher basal ATPase rate in detergent, however, the stimulation effect is significantly less pronounced, corresponding to only an ~50% increase for GSSG stimulation. Little change is observed for the $K_m$s of MgATP between the presence and absence of glutathione derivatives for either detergent solubilized or nanodisc reconstituted *At*Atm3 (*Figure 1*).

## Inward-facing, nucleotide-free conformational states

To map out the transport cycle, we attempted to capture *At*Atm3 in distinct liganded conformational states using single-particle cryoEM. We first determined the structure of *At*Atm3 reconstituted in nanodiscs at 3.4 Å resolution in the absence of either nucleotide or transport substrate (*Figure 2a* and *Figure 2—figure supplement 1*). This structure revealed an inward-facing conformation for *At*Atm3 similar to those observed for the inward-facing conformations for *Sc*Atm1 (PDB ID: 4myc) and *Na*Atm1 (PDB ID: 6vqu) with overall alignment root mean square deviations (rmsds) for the dimer of

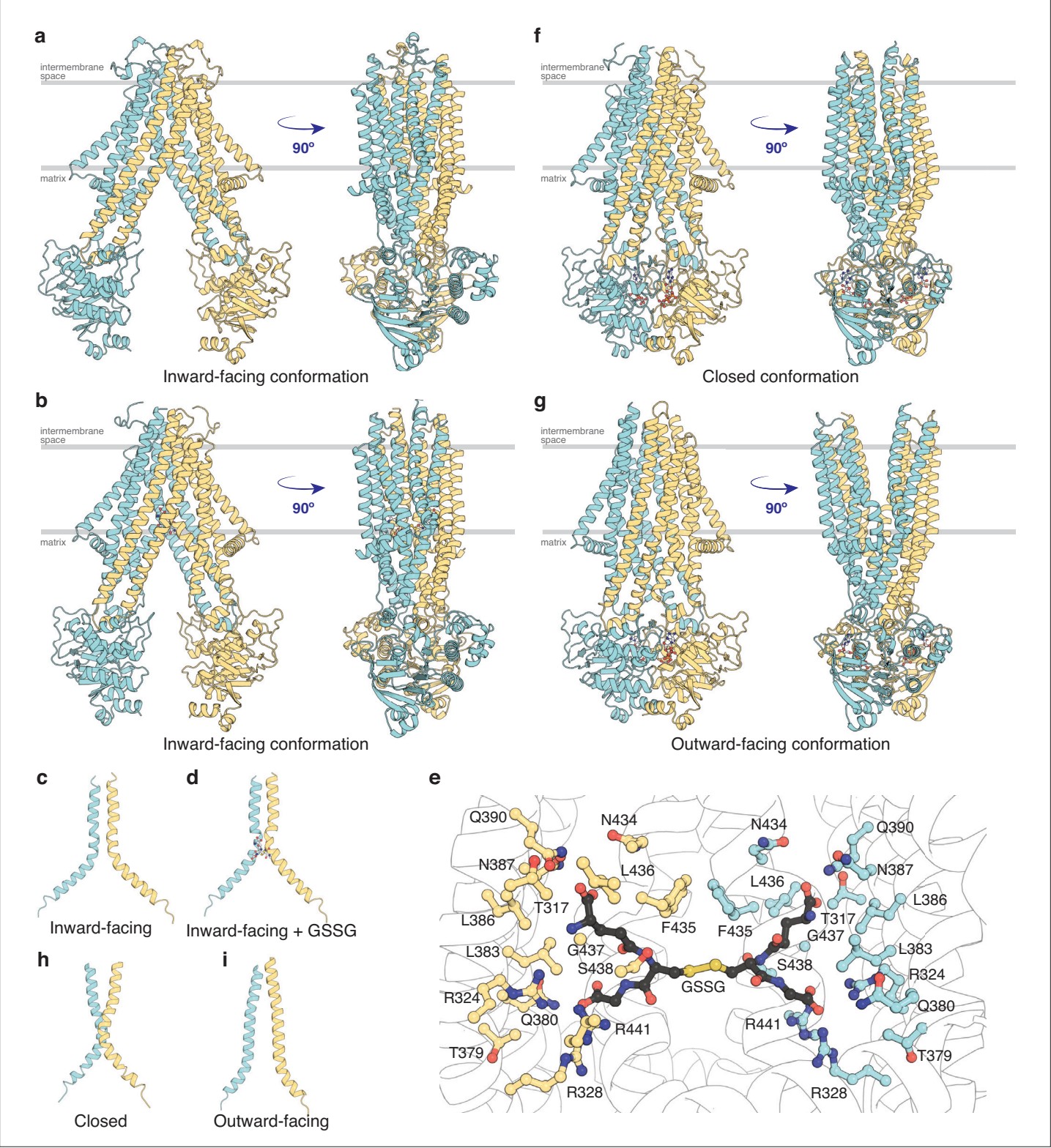

**Figure 2.** Structures of *At*Atm3. (**a**) Inward-facing conformation in the apo state. (**b**) Inward-facing conformation with oxidized glutathione (GSSG) bound. (**c**) TM6s (residues 416–460) in the inward-facing conformation. (**d**) TM6s in the GSSG-bound inward-facing conformation. The location of GSSG is indicated. (**e**) Residues important in stabilizing GSSG binding site, identified by PDBePISA (*Krissinel and Henrick, 2007*). (**f**) Closed conformation with MgADP-VO$_4$ bound. (**g**) Outward-facing conformation with MgADP-VO$_4$ bound. (**h**) TM6s in the closed conformation. (**i**) TM6s in the outward-facing conformation.

*Figure 2 continued on next page*

*Figure 2 continued*

The online version of this article includes the following figure supplement(s) for figure 2:

**Figure supplement 1.** Single-particle cryo-electron microscopy (cryoEM) structure of *At*Atm3 in the inward-facing conformation.

**Figure supplement 2.** Structural alignment of *At*Atm3 to other ATM transporters.

**Figure supplement 3.** Half-transporter comparison of transporters in the ATM family.

**Figure supplement 4.** Single-particle cryo-electron microscopy (cryoEM) structure of *At*Atm3 in the inward-facing conformation with oxidized glutathione (GSSG) bound.

**Figure supplement 5.** Structural alignment of *At*Atm3 in the inward-facing conformation.

**Figure supplement 6.** Single-particle cryo-electron microscopy (cryoEM) structure of *At*Atm3 in the closed conformation.

**Figure supplement 7.** Single-particle cryo-electron microscopy (cryoEM) structures of *At*Atm3 in the outward-facing conformation.

**Figure supplement 8.** Structural alignment of *At*Atm3 in the closed and outward-facing conformation.

2.6 Å (*Figure 2—figure supplement 2a, b*) and 2.1 Å (*Figure 2*), respectively, and half-transporter alignment rmsds of 2.3 and 2.0 Å (*Figure 2—figure supplement 2c*), respectively. The primary distinction between these structures is the presence of an approximately 20 amino acid loop between TM1 and TM2 of *At*Atm3 that would be positioned in the intermembrane space and is absent from the structures of ABCB7 (*Jumper et al., 2021*; *Varadi et al., 2022*), ABCB6 (*Song et al., 2021*), *Sc*Atm1 (*Srinivasan et al., 2014*), and *Na*Atm1 (*Lee et al., 2014*; *Figure 2—figure supplement 3*). While the functional relevance of this loop in *At*Atm3 is not known, structural characterization of PglK, a lipid-linked oligosaccharide flippase, revealed a comparably positioned external helix between TM1 and TM2 that was implicated in substrate flipping (*Perez et al., 2015*), suggestive that the corresponding loop could also have a functional or structural role in *At*Atm3.

To further look at substrate binding, we determined a 3.6 Å resolution single-particle cryoEM structure of *At*Atm3 purified in DDM with bound GSSG (*Figure 2b* and *Figure 2—figure supplement 4*). Although the overall resolution of the reconstruction was moderate (*Figure 2—figure supplement 4d*), we were able to model the GSSG molecule into the density map. In this structure, *At*Atm3 adopts an inward-facing conformation, with an overall alignment rmsd to the ligand-free inward-facing structure of 2.9 Å (*Figure 2—figure supplement 5a*) and a corresponding half-transporter alignment rmsd of 1.6 Å (*Figure 2—figure supplement 5b*). The main difference between the two structures is the extent of NBD dimer separation (*Figure 2—figure supplement 5a*), where the GSSG-bound structure presents a more closed NBD dimer relative to the substrate-free structure. As previously noted with *Na*Atm1 (*Fan et al., 2020*), the TM6s in these inward-facing structures of *At*Atm3 adopt a kinked conformation including residues 429–438 (*Figure 2cd*). This opens the backbone hydrogen bonding interactions to create the binding site for GSSG (*Figure 2e*) with binding pocket residues identified by PDBePISA (*Krissinel and Henrick, 2007*). The binding mode of GSSG in this *At*Atm3 inward-facing conformation is similar to that observed in the inward-facing structure of the GSSG-bound *Na*Atm1 (*Lee et al., 2014*).

## MgADP-VO$_4$ stabilized closed and outward-facing conformational states

MgADP-VO$_4$ has been found to be a potent inhibitor of multiple ATPases through formation of a stable species resembling an intermediate state during ATP hydrolysis (*Crans et al., 2004*; *Davies and Hol, 2004*). We determined two structures of *At*Atm3 stabilized with MgADP-VO$_4$, one in the closed conformation with *At*Atm3 reconstituted in nanodiscs at 3.9 Å resolution (*Figure 2f* and *Figure 2—figure supplement 6*) and the other in the outward-facing conformation with *At*Atm3 in DDM at 3.8 Å resolution (*Figure 2g* and *Figure 2—figure supplement 7*). These two structures share an overall alignment rmsd of 1.7 Å with the primary difference being a change in separation of the TMs surrounding the translocation pathway on the side of the transporter facing the intermembrane space (*Figure 2—figure supplement 8*). As a result of these changes in the TMDs, access to the intermembrane space is either blocked in the closed conformation (*Figure 2f*) or is open in the outward-facing conformation (*Figure 2g*). The changes in the TMDs are reflected in the conformations of TM6, which in the closed structure presents a kinked conformation (*Figure 2h*), in contrast to the straight conformation in the outward-facing structure that has the backbone hydrogen bonding interaction restored in the helices (*Figure 2i*). Further, the loops between TM1 and TM2 that are characteristics

of the *At*Atm3 transporter are better ordered in the closed conformation than in the outward-facing conformation (*Figure 2fg* and *Figure 2—figure supplements 6 and 7*). In contrast to the variation in the TMDs, the dimerized NBDs are virtually identical in these two structures with an overall alignment rmsd of 0.8 Å (*Figure 2fg* and *Figure 2—figure supplement 8*).

## Discussion

The plant mitochondrial Atm3 transporter has been implicated in a diverse set of functions associated with transition metal homeostasis that are reflective of the roles that have been described for the broader Atm1 transporter family. To provide a general framework for addressing the detailed function of this transporter in plants, we have structurally and functionally characterized Atm3 from *Arabidopsis thaliana*. We first identified a construct of *At*Atm3 with the mitochondrial targeting sequence deleted that expressed well in *E. coli* (*Figure 1—figure supplement 1*). Following purification, the ATPase activities of *At*Atm3 were measured in both detergent and MSP nanodiscs as a function of MgATP concentrations (*Figure 1*). Overall, the ATPase rate measured in detergent is about fivefold greater than that measured in nanodiscs, perhaps indicative of a more tightly coupled ATPase activity in a membrane-like environment. Both GSH and GSSG stimulate the ATPase activity by increasing $V_{max}$, with little change observed in the $K_m$ for MgATP. The ability of GSSG to stimulate the ATPase activity of *At*Atm3 agrees with previous reports (*Schaedler et al., 2014*), while the stimulation we observe with 10 mM GSH differs from the lack of stimulation noted in that work with 1.7 and 3.3 mM GSH. This discrepancy may reflect the higher GSH concentration utilized in the present studies, as well as differences in other experimental conditions including the use of a *Lactococcus lactis* expression system and Δ60 N-terminal truncation by *Schaedler et al., 2014*, compared to the *E. coli* expression system and the Δ80 N-terminal truncation employed in the present work.

ABC transporters are typically envisioned as utilizing an 'alternating access' mechanism, in which the substrate binding site transitions between inward- and outward-facing conformations coupled to the binding and hydrolysis of ATP. In an idealized two-state model, ABC transporters only adopt these two limiting conformations, but structural characterizations of ABC transporters in the presence of nucleotides and substrate analogs have identified a variety of intermediates, including occluded (with a ligand binding cavity exhibiting little or no access to either side of the membrane) and closed (no ligand binding cavity) conformations. The most extensive analysis of the conformational states of an Atm1-type exporter has been detailed for *Na*Atm1 and assigned to various states in the transport cycle (*Fan et al., 2020*; *Lee et al., 2014*). In the present work, we have determined four structures of *At*Atm3 in three different conformational states by single-particle cryoEM: two inward-facing conformations (with and without bound GSSG) (*Figure 2ab*), together with closed and outward-facing states stabilized by MgADP-VO$_4$ (*Figure 2fg*). The parallels between the structurally characterized conformations of *At*Atm3 and *Na*Atm1 support the idea that these conformational states are relevant to the transport cycle, and not simply an artifact of the specific conditions used to prepare each sample. The conformations observed for *At*Atm3 and *Na*Atm1 do not completely correspond, however; most notably, the outward-facing conformation observed for *At*Atm3 had not been previously observed with *Na*Atm1 (*Fan et al., 2020*; *Lee et al., 2014*), while the occluded conformations found with *Na*Atm1 were not observed for *At*Atm3.

The closed conformation stabilized by MgADP-VO$_4$ has been observed for both *Na*Atm1 (*Fan et al., 2020*) and *At*Atm3. In comparing the closed and the outward-facing conformations of *At*Atm3, the arrangements of the NBDs are superimposable, with the major differences between the two conformational states involving the local conformation of TM6s. In the closed conformation, the TM6s adopt a kinked conformation at residues 438–441 that eliminates the substrate binding cavity, whereas the TM6s in the outward-facing conformation present straight TM6s (*Figure 2hi*). The TM6 helical kink in the closed conformation is adjacent to, but distinct from, the helical kink present at residues 429–438 in the inward-facing conformation. By eliminating the substrate binding cavity, the presence of the closed conformation in a post-ATP hydrolysis state enforces unidirectionality of the transport process by precluding the uptake of substrate from the outside. We note that MgADP-VO$_4$ was observed to stabilize two different conformational states for *At*Atm3, the closed state in nanodiscs and the outward-facing conformation in detergent (*Figure 2fg*). The stabilization of multiple conformations of *At*Atm3 with MgADP-VO$_4$ contrasts with our previous observations on *Na*Atm1, where only the closed conformation was observed (*Fan et al., 2020*). The underlying basis for these differences is not known but may reflect differences in the conformational stabilization of the membrane spanning

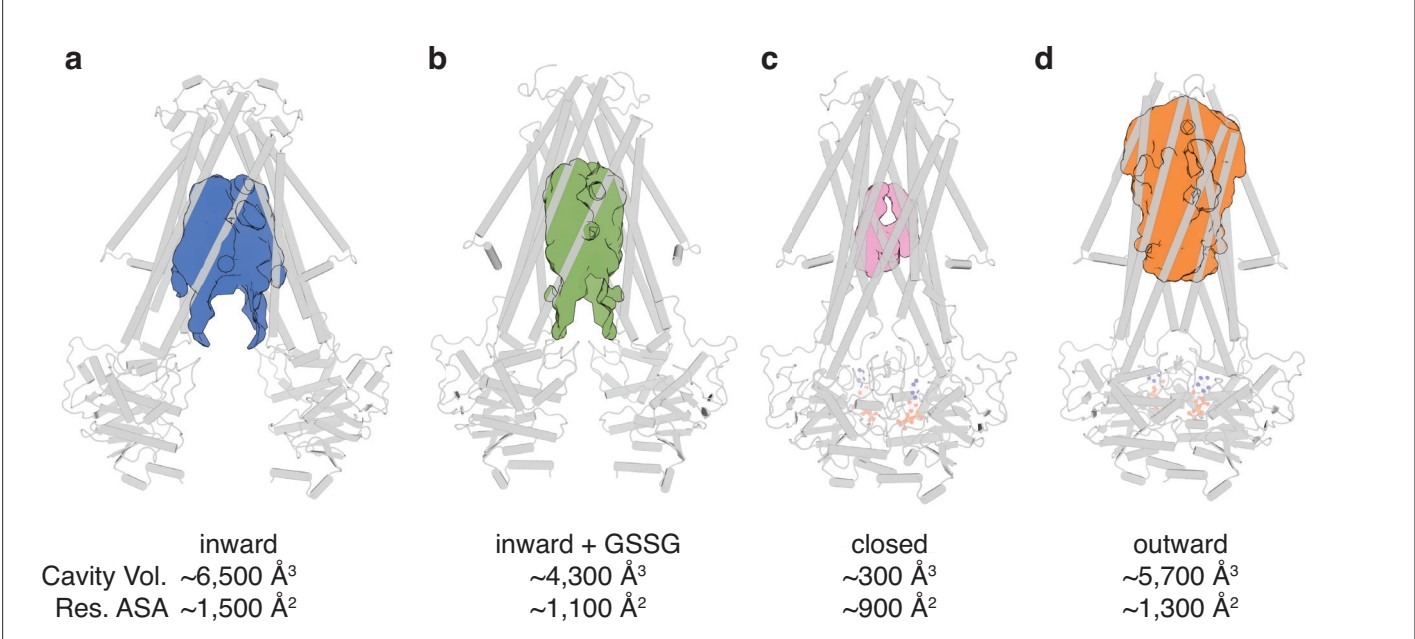

**Figure 3.** Binding cavity analysis. (**a**) Central cavity of the apo inward-facing conformation. (**b**) Central cavity of the inward-facing conformation with oxidized glutathione (GSSG) bound. (**c**) Closed conformation with a restricted cavity observed. (**d**) Central cavity of the outward-facing conformation. Cavity volumes were measured by CastP (*Tian et al., 2018*) using a probe radius of 2.5 Å. *At*Atm3 is shown as a gray cartoon representation, while cavities are depicted as color surfaces. The accessible solvent areas (ASA) of the key residues in the GSSG binding pockets of different structures were calculated by Areaimol in CCP4 (*Winn et al., 2011*).

The online version of this article includes the following figure supplement(s) for figure 3:

**Figure supplement 1.** Accessible surface area of binding site residues.

regions between detergents and nanodiscs. Structural differences between detergents and nanodiscs have been previously reported for MsbA under MgADP-VO₄ stabilizing conditions (*Mi et al., 2017*; *Ward et al., 2007*), and in the functional analysis of other membrane proteins (*Hänelt et al., 2013*).

In contrast to differences in the structures of the TMDs between the closed and outward-facing conformations, the TMDs in the inward-facing conformation structures of *At*Atm3 are similar. The primary differences between the two structures of inward-facing conformations of *At*Atm3 are in the relative positioning of the NBDs which are more widely separated in the apo structure relative to the GSSG-bound structure. Similar substrate-induced NBD movements have been previously observed in MRP1 (*Johnson and Chen, 2017*) and ABCB1 (*Barbieri et al., 2021*).

The conformational changes in the TMDs underlying the transport cycle are associated with changes in the extent of kinking of TM6 and the positioning of TM4-TM5 relative to the core formed by the remaining four TMs. As noted for *Na*Atm1, we observed kinked TM6s in the inward-facing and closed state of *At*Atm3 (*Figure 2cdh*), but not the outward-facing conformation (*Figure 2i*). These conformational changes lead to changes in the volume of the central cavity forming the gluta-thione binding site. Using the program CastP (*Tian et al., 2018*) with a probe radius of 2.5 Å, the cavity volumes of the inward-facing apo and GSSG-bound structures were measured to be ~6500 Å³ (*Figure 3a*) and ~4300 Å³ (*Figure 3b*), respectively, while the closed conformation exhibits a cavity volume of ~300 Å³ (*Figure 3c*), and the outward-facing conformation has a cavity volume of ~5700 Å³ (*Figure 3d*). We also measured the accessible solvent areas (ASA) of the key residues forming the binding site for GSSG in the different conformational states using Areaimol in CCP4 (*Winn et al., 2011*); the ASA of the inward-facing, inward-facing with GSSG bound, closed, and outward-facing structures are ~1500, ~1100, ~900, and ~1300 Å², which are also highly correlated with the cavity volume calculations. Most of the binding pocket residues remain exposed in all conformations with a few having large relative changes than others (*Figure 3—figure supplement 1*). Further, the cavity volume measurements are comparable to those calculated for *Na*Atm1 (*Fan et al., 2020*). The simi-larities in conformational states between *Na*Atm1 and *At*Atm3 indicate these transporters follow

the same basic mechanism, in which straightening of TM6s in the transition from inward to outward conformation leads to the release of substrate to the opposite side of the membrane. Following substrate release, the transporter resets to the inward-facing conformation through the closed conformation adopted after ATP hydrolysis; the decreased size of the substrate binding cavity helps enforce substrate release and unidirectionality of substrate transport.

The binding pocket for GSSG identified in this work primarily consists of residues from TM5 and TM6, with additional contributions from residues in TM3 and TM4 (*Figure 4—figure supplement 1*). The GSSG binding site for *At*Atm3 largely overlaps with that identified previously for *Na*Atm1 (*Lee et al., 2014*) and for the binding of reduced GSH to *Sc*Atm1 (*Srinivasan et al., 2014*). Inspection of a sequence alignment of Atm1 homologs (*Figure 4—figure supplement 1*) reveals that those residues forming the glutathione binding site are largely conserved, particularly if they are involved in polar interactions. A striking feature is the stretch of residues from P432 to R441 in the middle of TM6 (*At*Atm3 sequence numbering) with sequence PLNFLGSVYR with a high degree of sequence conservation. P432 is associated with the TM6 kink in inward-facing conformations that permits formation of hydrogen bonds between exposed peptide groups with GSSG (*Lee et al., 2014*); as TM6 straightens in the occluded and outward-facing conformations, these peptide groups are no longer available to bind the transport substrate (*Fan et al., 2020*). A sequence alignment of the structurally characterized *At*Atm3, *Na*Atm1, *Sc*Atm1, and human ABCB7 and ABCB6 transporters establishes that residues in the binding pockets are conserved, including T317, R324, R328, N387, Q390, L433, G437, and R441 (*Figure 4—figure supplement 1*). The conservation of binding pocket residues as calculated by the program ConSurf (*Landau et al., 2005*) is illustrated in *Figure 4* suggests that the substrates for these transporters may share common features, such as the glutathione backbone. Positions such where sequence variability is evident, such as residue 435 (*Figure 4b* and *Figure 4—figure supplement 1*), may reflect the binding of distinct GSSG derivatives by different eukaryotic and prokaryotic homologs.

An important property of disulfide containing compounds such as GSSG is that they can undergo disulfide – thiol exchange with free -SH groups (*Creighton, 1984*; *Nagy, 2013*). This reactivity creates potential challenges for proteins such as *At*Atm3 since reaction of a disulfide containing ligand such as GSSG with the thiol-containing side chain of cysteine could lead to formation of the mixed disulfide, thereby covalently connecting glutathione to the protein and releasing reduced GSH. Formation of the covalently linked mixed disulfide would be expected to restrict the access of exogenous ligands to the substrate binding cavity and hence would inhibit transport. For membrane proteins, cysteine residues are present in TMs with a frequency of about 1% (*Baeza-Delgado et al., 2013*). Although no cysteines residues are present in the *At*Atm3 binding pocket, we analyzed additional Atm3 homologs from plants. For this analysis, we used the NCBI blastp server (*Altschul et al., 1997*) and selected 410 sequences with a sequence identity of 50–100% and query coverage of 80–100% with *At*Atm3. Within the six TMs, the overall presence of cysteines was found to be ~0.4%. In this alignment, no cysteines were found in residues forming the glutathione binding cavity (*Figure 5*); more strikingly, no cysteine residues were found at any position of TM6 for these homologs (*Figure 5f*). Cysteines that are present in the TMs are either distant from the binding site, such as position 405 in TM5 of *At*Atm3 (*Figure 5eg*) or if they are closer to the binding site, are positioned on the opposite side of the TMs, such as positions 149, 215, 290, and 307 (*Figure 5abcd*). The observed exclusion of cysteines from the glutathione binding site (*Figure 5g*) could consequently be the result of a selection against this residue to prevent the formation of inhibitory mixed disulfides during the transport cycle.

As a general strategy to stabilize ABC transporters in distinct conformational states, different nucleotides or transport substrates are mixed with the transporter. The expectation is that a particular set of ligands will stabilize a specific conformational state, and so we were surprised to have captured with MgADP-VO$_4$ both a closed conformation in MSP nanodiscs and an outward-facing conformation in detergent. Given the similarities in the NBDs between these two structures, the distinctive arrangements of the TMDs between the closed and outward-facing conformations presumably arise from differences in the TMD environment provided by MSP nanodiscs and DDM, respectively. Furthermore, this phenomenon of obtaining different conformational states with MgADP-VO$_4$ is not unprecedented, however, since previously determined structures of MgADP-VO$_4$ stabilized ABC exporters include *Na*Atm1 in the closed conformation (*Fan et al., 2020*), *Thermus thermophilus* TmrAB in the occluded and outward-facing conformations (*Hofmann et al., 2019*), and *E. coli* MsbA in the closed conformation (*Mi et al., 2017*).

Despite extensive efforts, we were unable to prepare the ternary complex of *At*Amt3 with both bound GSSG and MgATP. To assess more generally the relationship between the conformational states of ABC

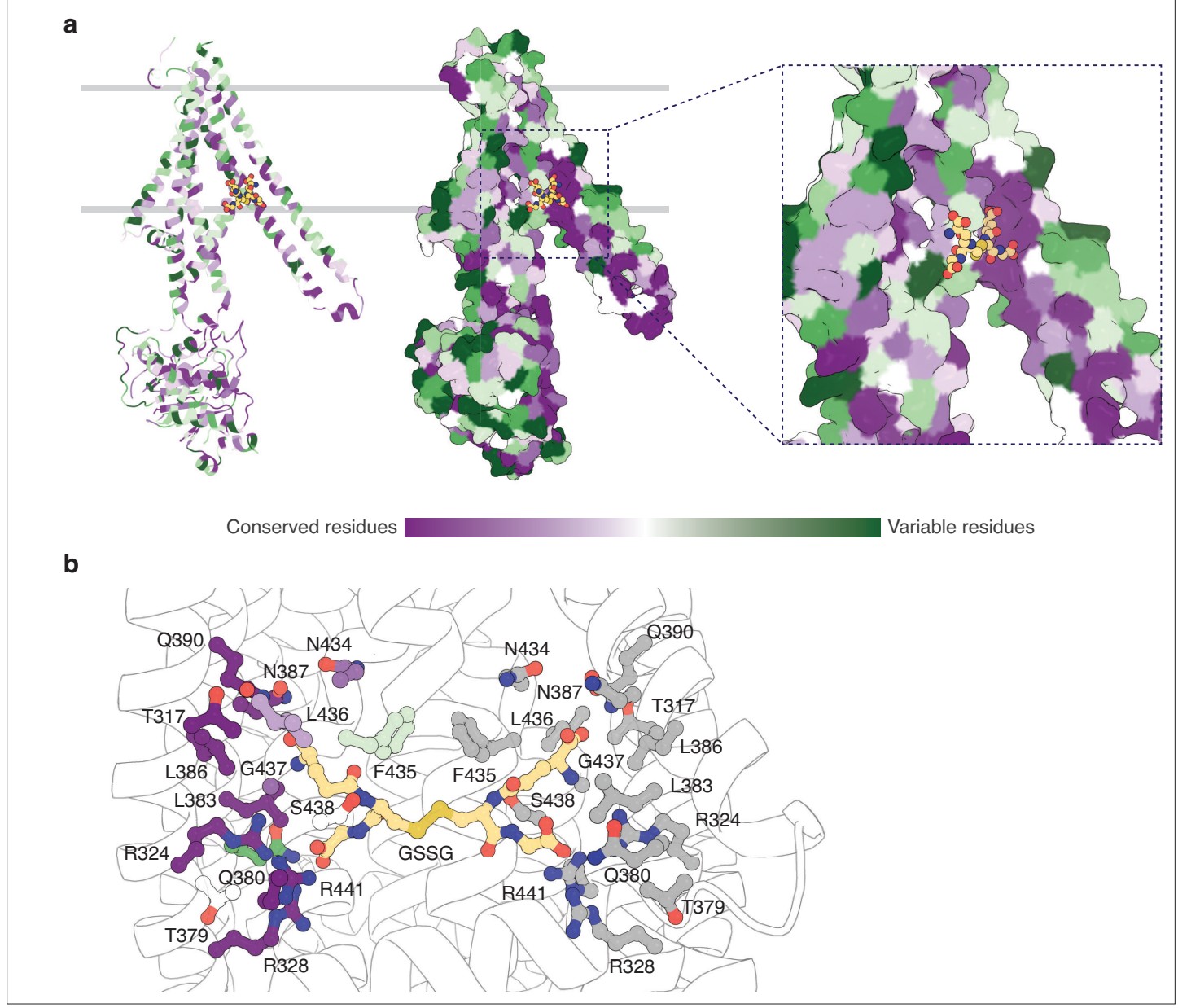

**Figure 4.** Substrate binding site conservation. (**a**) Sequence conservation of *At*Atm3, *Na*Atm1, *Sc*Atm1, human ABCB7, and human ABCB6 calculated by ConSurf (*Landau et al., 2005*) plotted on a cartoon and surface representations of a half-transporter of *At*Atm3 in the GSSG-bound inward-facing conformation. Oxidized glutathione (GSSG) is shown in spheres. (**b**) Conservation of key residues in the GSSG binding pocket. Residues in one chain colored based on the conservation and residues in the second chain are colored in gray. All residues and GSSG are shown in ball and sticks.

The online version of this article includes the following figure supplement(s) for figure 4:

**Figure supplement 1.** Sequence `alignment of selected Atm family transporters.

exporters and the presence or absence of nucleotide and transport substrate, we systematically compared the conformations of 80 half-transporters from the available structures of type IV ABC transporters (**Supplementary file 2**). To guide this analysis, the structures were mapped onto a one-dimensional conformational axis using principal component analysis (PCA; see Materials and methods). The PCA has the advantage for this purpose of separating all structures along a single axis such that transporters with similar structures will generally be positioned more closely together. Since only the dominant component is used, however, this analysis simplifies the conformational richness of ABC transporters. We note that the PCA does not provide a unique measure of the conformational state, and in particular, alternative metrics could be employed such

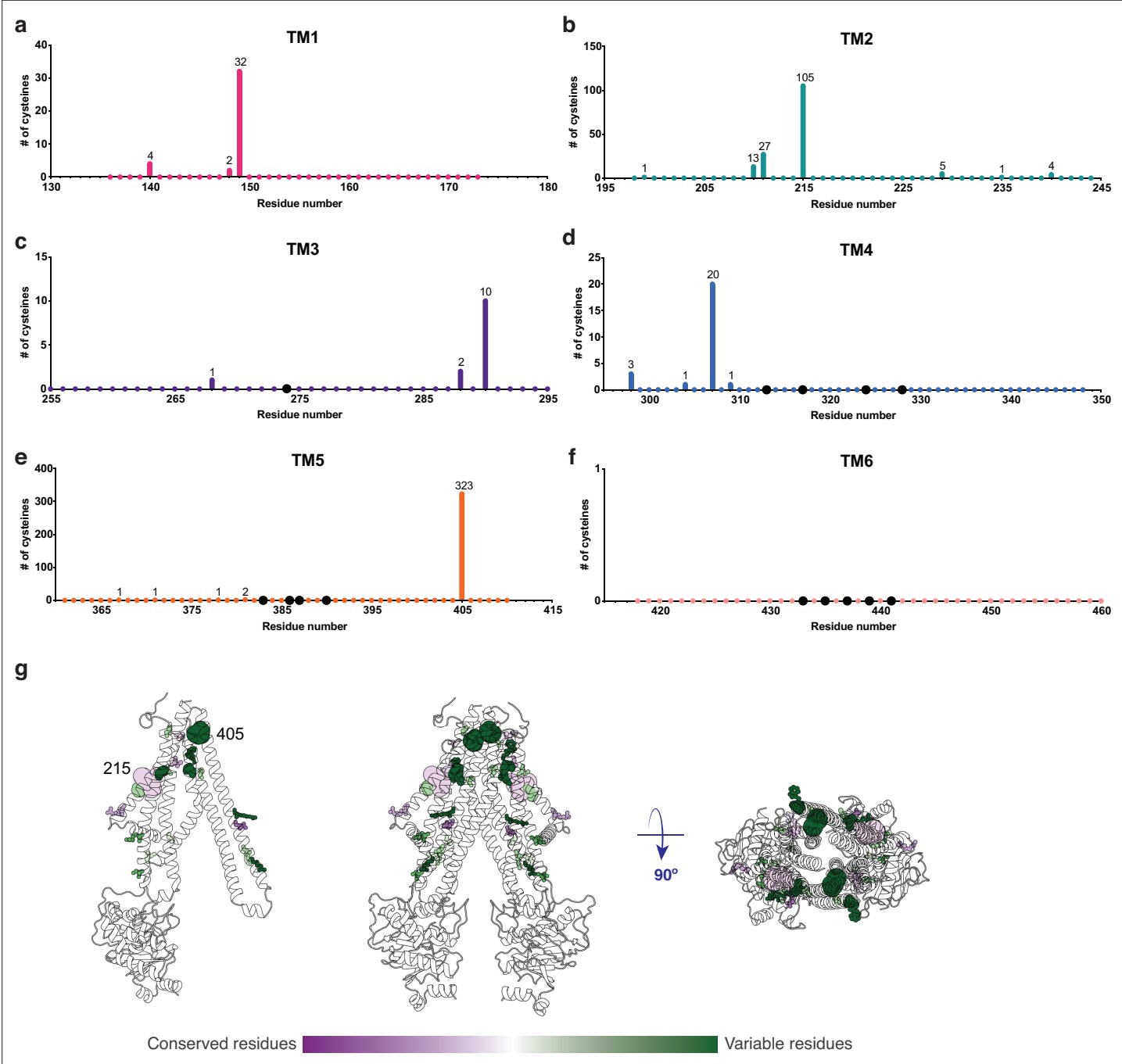

**Figure 5.** Cysteine residues found in transmembrane helices in an alignment of 410 *At*Atm3 homologs. (**a–f**) Residues are numbered based on the *At*Atm3 sequence. Small colored dots represent different residue positions, while the larger black dots indicate residues observed to interact with glutathione in the *Na*Atm1 and *At*Atm3 structures. The numbers above a given residue indicate the number of sequences in the alignment with a cysteine at that position; unlabeled positions denote positions where no cysteines were observed in the alignment. (**g**) Transmembrane cysteine residues in the inward-facing conformation of *At*Atm3. All positions with cysteine counts are shown in spheres. *At*Atm3 residues at positions that have 1–10 counts of cysteines are depicted with small spheres, residues at positions that have 11–100 counts of cysteines are depicted with medium sized spheres, and residues at positions with 101 and greater counts of cysteines are depicted with large spheres. The spheres are colored based on the Consurf coloring used in *Figure 4*. The distance between the Cα's of two 215 residues is 37 Å, and the distance between the Cα's of two 405 residues is 25 Å.

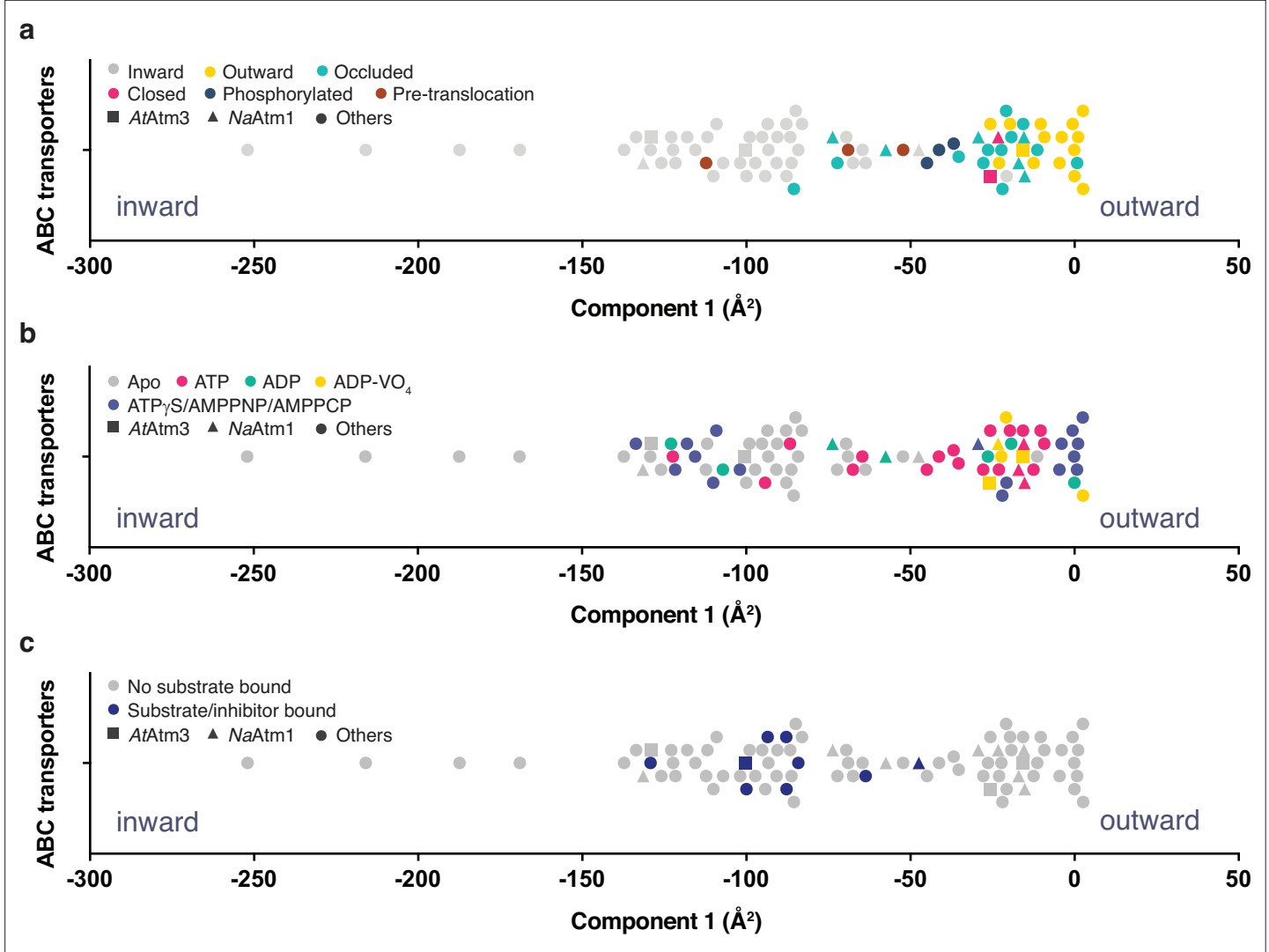

**Figure 6.** Principal component analysis (PCA) of type IV ATP binding cassette (ABC) exporters. (**a**) One-dimensional plot of component 1, colored according to the conformational states assigned in the original publications. The plot is oriented with the most inward and most outward conformations to the left and right, respectively. (**b**) One-dimensional plot of component 1 with inward-facing to outward-facing structures, colored by their nucleotide states. (**c**) One-dimensional plot of component 1 with inward-facing to outward-facing structures, colored by their substrate states. Each marker (square (■), triangle (▲), circle (●)) represents a unique half-transporter structure. Squares (■) represent structures of *At*Atm3, triangles (▲) represent structures of *Na*Atm1, and circles (●) represent structures for other ABC transporters.

The online version of this article includes the following figure supplement(s) for figure 6:

**Figure supplement 1.** Comparison of principal component analysis (PCA) and distance measurements of ATP binding cassette (ABC) exporters.

as measuring distances between particular pairs of residues. A comparison is presented in **Figure 6—figure supplement 1** comparing the dominant PCA component to alternative distance measures; these are generally anti-correlated, with the interesting exception of the distance between a pair of residues positioned in the outer facing (intermembrane space) that, not surprisingly, does a much better job of discriminating between outward conformations than the PCA, while less sensitive to differences between inward conformations. As the quantitative value of the principal component was utilized in this analysis only to separate structures along a single conformational analysis (**Figure 6**), we utilized the PCA for that purpose in this analysis.

The principal component is dominated by the conformational state of the TMDs, which represents ~62% of total conformational change from the inward-facing to the outward-facing conformation. The distribution of component 1 of the PCA is shown in **Figure 6** with the most extreme inward-facing conformations to the left and the outward-facing conformations to the right. To validate this approach, we color coded each

structure according to their published conformational state (*Figure 6a*). It is evident that outward-facing conformations occur on the right-handed side of the figure with component 1 values between –25 and 0 Å², while occluded, closed, and inward-facing conformations cluster around –25,–25, and –100 Å², respectively. Hence, the magnitude of the principal component does capture the expected trends in conformational state, with increasing values corresponding to a progression from inward-facing to outward-facing conformations. The correspondence is not exact, however, since assigned conformational states overlap, which could reflect either limitations of the PCA and/or inconsistent assignments of conformational states between different structures. We further note that some conformational states have a very wide distribution, particularly the inward-facing conformations, with several structures exhibiting widely separated subunits (with values for component 1 below –150 Å²).

Using the PCA, we could relate protein conformation to binding of nucleotides (*Figure 6b*) and substrates/inhibitors (*Figure 6c*). This analysis establishes that occluded and outward-facing conformations are largely nucleotide bound with either ATP or an ATP analog, but there are exceptions, most notably the original ADP-bound structure of Sav1866 in the outward facing conformation (*Dawson and Locher, 2006*; a similar Sav1866 structure was subsequently solved with bound AMPPNP; *Dawson and Locher, 2007*). Inward-facing conformations are observed in both nucleotide-free and nucleotide-bound forms (*Figure 6b*). Only the MgADP-VO₄-bound structures are exclusively found to occupy a small conformational space in the closed/outward-facing region. In contrast to the binding of nucleotides to all conformational states of these transporters, a distinct pattern is observed for the binding of transport substrates where the transporter invariably adopts the inward-facing conformation (*Figure 6c*). Intriguingly, no structures with associated (dimerized) NBDs to date have been published that contain both nucleotide and transport substrate, suggesting that this ternary complex exists only transiently during the transport cycle. As this is a key intermediate for understanding how the binding of transport substrate stimulates the ATPase activity, characterization of the ternary complex represents an outstanding gap in the mechanistic characterization of ABC exporters.

## Materials and methods
### Cloning, expression, and purification

A pET-21a (+) plasmid containing the full-length *A. thaliana* Atm3 (*At*Atm3) gene with a C-terminal 6x-His tag was purchased from Genscript (Genscript, Piscataway, NJ). Mutagenesis reactions generating the N-terminal 60, 70, and 80 amino acids deletion mutants were carried out with the Q5 mutagenesis kit (New England Biolabs, Ipswich, MA). All *At*Atm3 constructs were overexpressed in *E. coli* BL21-gold (DE3) cells (Agilent Technologies, Santa Clara, CA) using ZYM-5052 autoinduction media as described previously (*Fan et al., 2020*). Cells were harvested by centrifugation and stored at –80°C.

For purification, frozen cell pellets were resuspended in lysis buffer containing 100 mM NaCl, 20 mM Tris, pH 7.5, 40 mM imidazole, pH 7.5, 10 mM MgCl₂, and 5 mM β-mercaptoethanol (BME) in the presence of lysozyme, DNase, and cOmplete protease inhibitor tablet (Roche, Basel, Switzerland). The resuspended cells were lysed with an M-110L pneumatic microfluidizer (Microfluidics, Cambridge, MA). Unlysed cells and cell debris were removed by centrifugation at ~20,000 × *g* for 30 min at 4°C. The membrane fraction containing *At*Atm3 was collected by ultracentrifugation at ~113,000 × *g* for an hour at 4°C. The membrane fraction was then resuspended in buffer containing 100 mM NaCl, 20 mM Tris, pH 7.5, 40 mM imidazole, pH 7.5, and 5 mM BME and further solubilized by stirring with the addition of 1% DDM (Anatrace, Holland, OH) for an hour at 4°C. The DDM solubilized membrane was ultracentrifuged at ~113,000 × *g* for 45 min at 4°C to remove any unsolubilized material. The supernatant was loaded onto a prewashed NiNTA column. NiNTA wash buffer contained 100 mM NaCl, 20 mM Tris, pH 7.5, 50 mM imidazole, pH 7.5, 5 mM BME, and 0.02% DDM, while the elution buffer contained the same components, but with 380 mM imidazole. The eluent was subjected to size exclusion chromatography using HiLoad 16/60 Superdex 200 (GE Healthcare, Chicago, IL) with buffer containing 100 mM NaCl, 20 mM Tris, pH 7.5, 5 mM BME, and 0.02% DDM. Peak fractions were collected and concentrated to ~10 mg/ml using Amicon concentrators (Millipore, Danvers, MA).

## ATPase activity assay

ATPase assays were carried out as described previously for both the detergent purified and the reconstituted nanodisc samples at 25°C (*Fan et al., 2020*) using a molybdate-based colorimetric assay (*Chifflet et al., 1988*).

## Nanodisc reconstitution

For the *At*Atm3 structures in nanodiscs, the reconstitution was performed following the previously described protocol (*Fan et al., 2020*). The reconstitution was done with a 1:4:130 molar ratio of *At*Atm3: MSP1D1: POPC (Avanti Polar Lipids, Alabaster, AL). After overnight incubation at 4 °C, the samples were subjected to size exclusion chromatography with a Superdex 200 Increase 10/300 column (GE Healthcare, Chicago, IL). The peak fractions were directly used for grid preparation with the reconstituted samples at ~0.5 mg/ml. For the structure of *At*Atm3 with GSSG bound in the inward-facing conformation, the detergent purified protein was incubated with 10 mM GSSG, pH 7.5 at 4 °C for an hour with *At*Atm3 at 4 mg/ml before freezing grids.

## Grid preparation

For the *At*Atm3 structure with MgADP-VO$_4$ bound in the outward-facing conformation, the detergent purified protein was incubated with 4 mM ATP, pH 7.5, 4 mM MgCl$_2$, and 4 mM VO$_4^{3-}$ with protein at 5 mg/ml at 4 °C overnight before freezing grids. For all grids, 3 µL of protein solution was applied to freshly glow-discharged UltrAuFoil 2/2 200 mesh grids (apo inward-facing conformation and closed conformation, both in nanodiscs) and UltrAuFoil 1.2/1.3 300 mesh grids (Electron Microscopy Sciences, Hatfield, PA) (GSSG-bound inward-facing conformation and outward-facing conformation, both in detergent) and blotted for 4–5 s with a blot force of 0% and 100% humidity at room temperature using the VitroBot Mark IV (Thermo Fisher, Waltham, MA).

## Single-particle cryoEM data collection, processing, and refinement

Datasets for the inward-facing conformations in apo and GSSG-bound states, and the outward-facing conformation with MgADP-VO$_4$ bound were collected with a Gatan K3 direct electron detector (Gatan, Pleasanton, CA) on a 300 keV Titan Krios (Thermo Fisher, Waltham, MA) in the super-resolution mode using SerialEM at the Caltech CryoEM facility. These datasets were collected using a defocus range between –1.5 and –3.0 µm and a total dosage of 60 e$^-$/Å$^2$. The dataset for the closed conformation with MgADP-VO$_4$ bound was collected with a Falcon 4 direction electron detector (Thermo Fisher, Waltham, MA) on a 300 keV Titan Krios (Thermo Fisher, Waltham, MA) in the super-resolution mode using EPU (Thermo Fisher, Waltham, MA) at the Stanford-SLAC Cryo-EM Center (S2C2) with a defocus range between –1.5 and –2.1 µm and a total dosage of ~48 e$^-$/Å$^2$.

Detailed processing workflows of all single-particle cryoEM datasets are included in *Figure 1—figure supplement 1*, *Figure 2—figure supplement 4* , *Figure 2—figure supplements 6 and 7*. with data collection, refinement, and validation statistics presented in *Supplementary file 1*. Datasets for the inward-facing conformation in apo and GSSG-bound states, and the outward-facing conformation with MgADP-VO$_4$ bound were motion corrected with the patch motion correction in cryoSPARC 2 (*Punjani et al., 2017*), while the dataset for the closed conformation with MgADP-VO$_4$ bound was motion corrected with motioncor2 (*Zheng et al., 2017*). The subsequent processing of all datasets was performed in a similar fashion. The contrast transfer function (CTF) parameters were estimated with patch CTF estimation in cryoSPARC 2 (*Punjani et al., 2017*). Particles were picked with blob picker using a particle diameter of 80–160 Å and then extracted. Rounds of two-dimensional and three-dimensional classifications were performed, leaving 157,762, 259,020, 140,569, and 103,161 particles for the inward-facing apo, inward-facing GSSG bound, closed, and outward-facing conformations, respectively. The final reconstructions were refined with homogeneous, non-uniform, and local refinements in cryoSPARC 2 with C2 symmetry (*Punjani et al., 2017*). The masks used in local refinements were generated in Chimera (*Pettersen et al., 2004*).

The initial model of the *At*Atm3 inward-facing conformation in apo state was obtained using the inward-facing occluded structure of *Na*Atm1 (PDB ID: 6pam) as a starting model (*Fan et al., 2020*). The model fitting was carried out with phenix.dock_in_map (*Liebschner et al., 2019*). That apo inward-facing conformation model of *At*Atm3 was subsequently used as the starting model for the inward-facing GSSG-bound structure. The previous *Na*Atm1 closed conformation structure (PDB: 6par) was

used as the starting model for both the closed and the outward-facing conformations. Model building and ligand fitting were carried out manually in *Coot* (*Emsley et al., 2010*) and the structures were refined with phenix.real_space_refine (*Liebschner et al., 2019*).

## Structure superposition

Structure superpositions for calculating the rmsds between different structures were performed with the SSM option in *Coot* (*Emsley et al., 2010*).

## Principal component analysis

The objective of our PCA was to distribute structures along a single conformational axis to correlate the binding of nucleotides and transported ligands to the conformational state of type IV ABC transporters that include *At*Atm3 and related Atm1 transporters (*Thomas et al., 2020*). For this purpose, we first identified 10 polypeptide stretches containing 7–21 residues at equivalent positions in 80 structurally characterized type IV half-transporters (*Supplementary file 2*), including residues from the TMD and from the NBD. In this sequence selection process, a single half-transporter was used for homodimeric transporters, and both half-transporters were used for heterodimeric transporters, whether encoded by two different half-transporter peptides or on a single peptide. The coordinates of Cα positions for the selected residues were extracted and aligned to that of the outward-facing conformation of Sav1866 (PDB ID: 2hyd) based on the Cα coordinates in TM3 and TM6, which were previously found to provide a useful reference frame for studying conformational changes (*Lee et al., 2014*). *At*Atm3 residues used in alignment: 140–160, 225–245, 255–275, 322–342, 362–382, 423–443, 504–513, 517–524, 618–632, and 681–688. *Na*Atm1 residues used in alignment: 36–56, 107–127, 137–157, 204–224, 244–264, 305–325, 386–395, 399–406, 500–514, and 563–570.

The PCA was performed using the 'essential dynamics' algorithm (*Amadei et al., 1993*). The full transporter was used in these calculations with the outward-facing state of Sav1866 (PDB ID: 2hyd) serving as the reference state. The first component captured 62% of the overall conformational variation among these structures, and so the eigenvalues corresponding to this component were used to order the different structures along a single axis (*Figure 6*). In general, the conformational states assigned to each structure parallel those obtained from the PCA; differences likely reflect the absence of standardized definitions for assigning the conformational states of ABC transporters as well as the limitations of this PCA, particularly the use of only the dominant eigenvector. A comparison of the PCA to alternative metrics based on the conformationally sensitive distances between specific pairs of residues is illustrated in *Figure 6—figure supplement 1*, with corresponding distances tabulated in *Supplementary file 2*.

# Acknowledgements

We thank Andrey Malyutin, Songye Chen, and Corey Hecksel for their support during single-particle cryoEM data collections and the reviewers for their helpful comments, particularly concerning comparisons of *At*Atm3 with other ABC transporters. CryoEM was performed in the Beckman Institute Resource Center for cryo-Electron Microscopy at Caltech and at the Stanford-SLAC Cryo-EM Center (S2C2). The S2C2 is supported by the National Institutes of Health Common Fund Transformative High Resolution Cryo-Electron Microscopy program. We thank the Beckman Institute for their support of the cryoEM facility at Caltech. Funding: DCR is a Howard Hughes Medical Institute Investigator.

# Additional information

## Funding

| Funder | Grant reference number | Author |
| --- | --- | --- |
| Howard Hughes Medical Institute | | Douglas C Rees |

The funders had no role in study design, data collection and interpretation, or the decision to submit the work for publication.

## Author contributions
Chengcheng Fan, Conceptualization, Data curation, Formal analysis, Investigation, Methodology, Validation, Visualization, Writing – original draft, Writing – review and editing; Douglas C Rees, Conceptualization, Funding acquisition, Methodology, Project administration, Software, Supervision, Writing – review and editing

## Author ORCIDs
Chengcheng Fan  http://orcid.org/0000-0003-4213-5758
Douglas C Rees  http://orcid.org/0000-0003-4073-1185

## Decision letter and Author response
Decision letter https://doi.org/10.7554/eLife.76140.sa1
Author response https://doi.org/10.7554/eLife.76140.sa2

# Additional files

## Supplementary files
• Supplementary file 1. Cryo-electron microscopy (cryoEM) data collection, refinement, and validation statistics.

• Supplementary file 2. Principal component analysis (PCA) and distance measurements. Calculated component 1 values in PCA are listed for different transporters. For heterologous transporters, transporter encoded in one polypeptide and transporters with different conformational states in one PDB file, '_A/B' is added at the end of each PDB to represent different half-transporters and/or different conformations. Residues used for distance measurements and the corresponding distance for each transporter are listed. Grayed out cells represent distances that cannot be measured for the second half-transporters of heterodimeric transporters.

• Transparent reporting form

• Source code 1. Fortran source code for the Principal Component Analysis.

## Data availability
The atomic coordinates for inward-facing, inward-facing with GSSG bound, closed and outward-facing conformations were separately deposited in the Protein Data Bank (PDB) and the Electron Microscopy Data Bank (EMDB) with accession codes: PDB 7N58, 7N59, 7N5A and 7N5B; EMDB EMD-24182, EMD-24183, EMD-24184 and EMD-24185. The plasmids encoding full-length AtAtm3 and the AtAtm3 with N-terminal 80 residue deletion were deposited in Addgene with Addgene ID 172321 and 173045, respectively. The raw data for ATPase assays presented in Figure 1 are provided in Source Data 1, while the essdyn.f Fortran source code used for the PCA analysis is provided as Source Code 1.

The following datasets were generated:

| Author(s) | Year | Dataset title | Dataset URL | Database and Identifier |
|---|---|---|---|---|
| Fan C, Rees DC, Drew D | 2022 | Glutathione binding to the plant AtAtm3 transporter and implications for the conformational coupling of ABC transporters | https://www.rcsb.org/structure/7N58 | RCSB Protein Data Bank, 7N58 |
| Fan C, Rees DC, Drew D | 2022 | Glutathione binding to the plant AtAtm3 transporter and implications for the conformational coupling of ABC transporters | https://www.rcsb.org/structure/7N59 | RCSB Protein Data Bank, 7N59 |

*Continued on next page*

*Continued*

| Author(s) | Year | Dataset title | Dataset URL | Database and Identifier |
|---|---|---|---|---|
| Fan C, Rees DC, Drew D | 2022 | Glutathione binding to the plant AtAtm3 transporter and implications for the conformational coupling of ABC transporters | https://www.rcsb.org/structure/7N5A | RCSB Protein Data Bank, 7N5A |
| Fan C, Rees DC, Drew D | 2022 | Glutathione binding to the plant AtAtm3 transporter and implications for the conformational coupling of ABC transporters | https://www.rcsb.org/structure/7N5B | RCSB Protein Data Bank, 7N5B |
| Fan C, Rees DC, Drew D | 2022 | Glutathione binding to the plant AtAtm3 transporter and implications for the conformational coupling of ABC transporters | http://www.ebi.ac.uk/pdbe/entry/emdb/EMD-24182 | Electron Microscopy Data Bank, EMD-24182 |
| Fan C, Rees DC, Drew D | 2022 | Glutathione binding to the plant AtAtm3 transporter and implications for the conformational coupling of ABC transporters | http://www.ebi.ac.uk/pdbe/entry/emdb/EMD-24183 | Electron Microscopy Data Bank, EMD-24183 |
| Fan C, Rees DC, Drew D | 2022 | Glutathione binding to the plant AtAtm3 transporter and implications for the conformational coupling of ABC transporters | http://www.ebi.ac.uk/pdbe/entry/emdb/EMD-24184 | Electron Microscopy Data Bank, EMD-24184 |
| Fan C, Rees DC, Drew D | 2022 | Glutathione binding to the plant AtAtm3 transporter and implications for the conformational coupling of ABC transporters | http://www.ebi.ac.uk/pdbe/entry/emdb/EMD-24185 | Electron Microscopy Data Bank, EMD-24185 |

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
