## [Editor Report]

Mitochondrial glutathione is an important line of defence against free radical production. The ATP binding cassette (ABC) transporter Atm3 exports oxidized glutathione out of the mitochondria to help maintain a suitable reducing environment. In this study, the authors have biochemically characterized Atm3 and determined four cryo-EM structures exhibiting three different conformational states, revealing new insights into the transport mechanism. This well-executed study will be of broad interest to the membrane biology and transport communities.

---

## [Decision Letter]

**Decision letter after peer review:**

Thank you for submitting your article "Glutathione binding to the plant *At*Atm3 transporter and implications for the conformational coupling of ABC transporters" for consideration by *eLife*. Your article has been reviewed by 3 peer reviewers, including David Drew as Reviewing Editor and Reviewer #1, and the evaluation has been overseen by Kenton Swartz as the Senior Editor. The following individual involved in review of your submission has agreed to reveal their identity: Markus A. Seeger (Reviewer #2).

Essential revisions:

1) We think that the closed conformation of Atm3 should be stressed further as it's only recently that a closed conformation was observed. This paper is another chance to announce/emphasize the existence of the closed conformation and the mechanistic conclusions that can be reached from it.

2) Please clarify the advantage of using principle component analysis (PC1) as compared to simply measuring distances between residues at strategic positions e.g. between coupling helices, at the extracellular gate and (to distinguish between closed and occluded) at the kinking prolines in TM6. One disadvantage with only PC1 is that one loses information on local changes described by PC2 etc. For example, we would assume that an inter-TM6 distance measurement is able to distinguish outward-occluded from closed; this being the most important advantage over the principal component analysis, but PC1 fails to distinguish outward-facing/outward-occluded and closed states here (as evident from Figure 6a). Some further discussion on this point is warranted and perhaps useful to clarify for those not familiar with using this statistical approach to clustering different conformational states.

3) The authors state in the abstract that "The absence of structures containing both nucleotide and transport substrate suggests that this ternary complex exists only transiently during the transport cycle." This provokes the question as to whether cryo-EM analyses were carried out in the presence of ATP and GSSG, i.e. under hydrolysis conditions and in the presence of the substrate? It is well established that outward-facing trapped ABC exporters (via vanadate trapping or EtoQ mutations at the Walker B motif) do not bind substrates (e.g. see MRP1 and ABCB1), hence it is not at all surprising that ATP-vanadate trapped AtAtm3 had no GSSG bound. However, the ternary complex may have been captured under hydrolysis conditions.

4) In Figure 1, the ATPase activities of AtAtm3 are quite different in detergent and membrane environment, with more than 10-fold decrease in basal activity from detergent to POPC containing nanodiscs. Please clarify if you think that the detergent ATPase signal is an over-estimate (e.g., due to increased dynamics without lipids present) or the nanodisc is an underestimate due to using POPC lipids rather than those found in the mitochondria membrane, e.g., requirement for cardiolipin? In one of the papers you cited, Schaedler et. al, 2014, ATM3's activity is not stimulated by GSH versus in your data, 10mM GSH strongly stimulated AtAtm3. How would you explain this discrepancy?

5) There are a couple of structural studies on Atm transporters available to date and the authors showed some structural comparisons in Figure S3 and S5. However, it is still not clear exactly what are new structural insights gained with these structures. It could be helpful to compare the substrate binding sites side-by-side or include a cartoon representation of different functional states in different systems. The authors brought readers attention to "a ~20 amino acid loop between TM1 and TM2 of AtAtm3 that would be positioned in the intermembrane space and is absent from the structures of ScAtm1 and NaAtm1". Does this loop have any proposed functional roles? Is it present in other Atm or ABC transporters?

6) The cysteine-exclusion hypothesis of Atm3 is interesting, but lacks experimental validation. Please provide some additional experimental data to support this (e.g., is a mixed disulfide formed between GSH and cysteine(s) introduced into binding site based on locations in structural homologues) and/or appropriately tone down the statement to more closely reflect the presented data.

*Reviewer #1 (Recommendations for the authors):*

The cysteine-exclusion of Atm3 would benefit from some additional experimental analysis, such as the introduction of cysteine residues as found in the closest structural homologues harbouring cysteines and their impact on oxidised glutathione export.

*Reviewer #2 (Recommendations for the authors):*

1) If this work is looked at in the larger context of type I ABC exporters (and in particular in the structure series of NaAtm1 published by the same authors in 2020 in PNAS), one burning question comes to my mind: How universal and generally relevant is what the authors call the closed conformation? In Fan et al., PNAS, 2020, the Rees lab for the first time (at least to the best of my knowledge) described a closed conformation for NaAtm1. The closed conformation has the NBDs fully dimerized/closed and the extracellular gate is closed, but in contrast to the more often observed outward-occluded conformation (first mentioning by Beis lab for McjD and ever since seen several times), there is no occluded cavity, but TM6-kinking results in a collapse of the internal cavity (thus called closed conformation). The major punch-line of the NaAtm1 story was that NaAtm1 assumes an outward-occluded conformation in the presence of non-hydrolysable AMP-PNP (pre-hydrolysis state), but a closed conformation in the presence of ATP-vanadate (post-hydrolysis state), suggesting a pumping/expelling mechanism that is dependent driven by ATP hydrolysis. Currently, the structural differentiation of AMP-PNP-bound versus ATP-vanadate bound state in light of the cavity collapse has only been described for NaAtm1 thus far, but not for any other type I ABC exporter. In line with the findings for NaAtm1, the authors describe in this paper the ATP-vanadate bound structure of AtAtm3 to assume again the closed conformation with kinked TMH6 and collapsed internal cavity (though, a small internal cavity still seems to exist according to Figure 3c). The obvious question (which does not appear to be mentioned in the paper) is whether the authors have attempted to obtain the AMP-PNP bound structure of AtAtm3 and if so, whether the outward-occluded conformation featuring an internal cavity was seen the way it was observed for NaAtm1?

Overall, I feel there is still limited appreciation in the ABC transporter field for the interesting and unique closed conformation, likely owing to the fact that the discovery of the first closed conformation is comparatively recent. This paper is another chance to announce/emphasize the existence of the closed conformation. Ideally, this is addressed by further structures (though that's only a reasonable request if it is straight-forward to achieve). Alternative possibilities are biochemical experiments (e.g. cross-linking, or DEER).

2) To stick to the "conformation business", I wonder whether the principal component analysis is superior to analyses simply measuring distances between residues at strategic positions (e.g. between coupling helices, at the extracellular gate and (to distinguish between closed and occluded) at the kinking prolines in TM6. Such distance measurements together with cavity determinations are in my view sufficient to distinguish between the relevant conformations and are more intuitive to grasp (see also PMID: 33098660).

In particular the distance between the coupling helices informs about NBD separation/completeness of NBD closure, hence allowing to differentiate between inward-facing/inward-occluded states (or the special case of the unlock return state of TmrAB) and outward-facing, outward-occluded and (as relevant here) the closed conformation. The closed conformation seems to correspond to an outward-occluded conformation when it comes the degree of intracellular/extracellular gate closure, but does not feature an enclosed cavity. I would assume that an inter-TM6 distance measurement is able to distinguish outward-occluded from closed; this being the most important advantage over the principal component analysis, which fails to distinguish outward-facing/outward-occluded and closed (as evident from Figure 6a). And finally, the degree of extracellular gate opening is of general interest when analysing different conformations with fully closed NBDs.

Unless I have overlooked something (which is well possible), I would consider such simple distance measurements as more informative than the principal component analysis.

3) The authors state in the abstract that "The absence of structures containing both nucleotide and transport substrate suggests that this ternary complex exists only transiently during the transport cycle." This provokes the question whether cryo-EM analyses were carried out in the presence of ATP and GSSG, i.e. under hydrolysis conditions and in the presence of the substrate. It is well established in the field that outward-facing trapped ABC exporters (via vanadate trapping or EtoQ mutations at the Walker B motif) do not bind substrates (e.g. see MRP1 and ABCB1), hence it is not at all surprising that ATP-vanadate trapped AtAtm3 had no GSSG bound. However, the ternary complex may have been captured under hydrolysis conditions.

*Reviewer #3 (Recommendations for the authors):*

1) In Figure 3, it could be helpful to label the functional states in panels in a, b, c, d, f, g, h. If the distances between 2 NBDs can be measured and labeled in panel a and b (apo and GSSG bound form), it would be more helpful for the non-ABC transporter audience to visualize the conformational change globally.

2) In Figure 6, the color codes in the legend do not match with those on the graph.

3) In Figure S4, there are 2 high-resolution reconstructions of AtAtm3 in an inward-facing conformation with GSSG bound (3.57A and 3.68A respectively), are they different structurally?

---

## [Author Response]

Essential revisions:1) We think that the closed conformation of Atm3 should be stressed further as it's only recently that a closed conformation was observed. This paper is another chance to announce/emphasize the existence of the closed conformation and the mechanistic conclusions that can be reached from it.

Thank you for this comment. We have amplified our discussion of the closed conformation on page 10. We also included a statement in the abstract of the different conformational states we observed for *At*Atm3, including the closed conformation.

2) Please clarify the advantage of using principle component analysis (PC1) as compared to simply measuring distances between residues at strategic positions e.g. between coupling helices, at the extracellular gate and (to distinguish between closed and occluded) at the kinking prolines in TM6. One disadvantage with only PC1 is that one loses information on local changes described by PC2 etc. For example, we would assume that an inter-TM6 distance measurement is able to distinguish outward-occluded from closed; this being the most important advantage over the principal component analysis, but PC1 fails to distinguish outward-facing/outward-occluded and closed states here (as evident from Figure 6a). Some further discussion on this point is warranted and perhaps useful to clarify for those not familiar with using this statistical approach to clustering different conformational states.

We utilized principal component analysis (PCA) to project the transporter structures onto a single conformational axis to distribute these structures so that similar structures will generally be positioned more closely together. Of course, since only the dominant component is used, this analysis simplifies the conformational richness of ABC transporters. We agree that the principal component is neither a completely accurate nor unique measure of conformational similarity, and in particular, alternative metrics could be employed such as measuring distances between particular pairs of residues. A comparison is presented in “Figure 6 supplement 1” comparing the dominant component to several alternative distance measures; these are generally anti-correlated, with the interesting exception of the distances between a pair of residues in the outer loops that, not surprisingly, do a much better job of discriminating between outward conformations than the PCA, while being less sensitive to differences between inward conformations. As the quantitative value of the principal component was only utilized in this analysis to separate structures along a single conformational analysis (Figure 6), we felt it was appropriate to use the PCA for that purpose in this analysis. This has been noted on page 14 of the text.

3) The authors state in the abstract that "The absence of structures containing both nucleotide and transport substrate suggests that this ternary complex exists only transiently during the transport cycle." This provokes the question as to whether cryo-EM analyses were carried out in the presence of ATP and GSSG, i.e. under hydrolysis conditions and in the presence of the substrate? It is well established that outward-facing trapped ABC exporters (via vanadate trapping or EtoQ mutations at the Walker B motif) do not bind substrates (e.g. see MRP1 and ABCB1), hence it is not at all surprising that ATP-vanadate trapped AtAtm3 had no GSSG bound. However, the ternary complex may have been captured under hydrolysis conditions.

We did try to capture this particular structure under turnover conditions in the presence of MgATP and the substrate GSSG, but unfortunately, we only observed the inward-facing conformation in these samples.

4) In Figure 1, the ATPase activities of AtAtm3 are quite different in detergent and membrane environment, with more than 10-fold decrease in basal activity from detergent to POPC containing nanodiscs. Please clarify if you think that the detergent ATPase signal is an over-estimate (e.g., due to increased dynamics without lipids present) or the nanodisc is an underestimate due to using POPC lipids rather than those found in the mitochondria membrane, e.g., requirement for cardiolipin? In one of the papers you cited, Schaedler et. al, 2014, ATM3's activity is not stimulated by GSH versus in your data, 10mM GSH strongly stimulated AtAtm3. How would you explain this discrepancy?

In our experience, the ATPase rates of ABC transporters are highly dependent on the precise solubilization conditions, particularly for detergents as some detergents/detergent combinations are stimulatory, while others are inhibitory. In addition, the POPC used in the nanodiscs might not be the best substitute of the natural lipids that are found in the mitochondrial membrane*.* Given the variability that has been observed in ATPase activities for different solubilization conditions, we do not know which condition more likely reflects the true physiological activity. (As an aside, we would have thought that basal ATPase rate should be zero in the absence of transported substrate to avoid uncoupled (and presumably wasteful) ATPase activity, but this has been rarely reported).

The Schaedler paper used GSH at two concentrations, 1.7 and 3.3 mM, to test the stimulation of ATPase activity. In our study, we used 10 mM GSSG for the ATPase activity assay to be consistent with our previous studies on *Na*Atm1. This discrepancy presumably reflects concentration differences utilized in these studies, and other experimental conditions including the nature of the protein construct. Different expression systems were used in the two studies, Schaedler et al., employed L. lactis, while we used *E. coli*; it is possible that different lipids co-purified with the proteins that could influence the ATPase activity. In addition, we used a truncation with an 80 residue deletion, while Schaedler et al., utilized a 60-residue truncation; our initial characterization of various constructs showed that the 80-residue deletion had a higher ATPase activity, although we did not characterize these differences in detail – certainly not for publication. While both studies used *At*Atm3 solubilized in DDM, it is possible that there are still differences in the solubilization conditions reflecting the precise way in which detergents are introduced. We have added a sentence on page 9 noting this discrepancy in the GSH stimulation results, without ascribing a particular mechanism.

5) There are a couple of structural studies on Atm transporters available to date and the authors showed some structural comparisons in Figure S3 and S5. However, it is still not clear exactly what are new structural insights gained with these structures. It could be helpful to compare the substrate binding sites side-by-side or include a cartoon representation of different functional states in different systems. The authors brought readers attention to "a ~20 amino acid loop between TM1 and TM2 of AtAtm3 that would be positioned in the intermembrane space and is absent from the structures of ScAtm1 and NaAtm1". Does this loop have any proposed functional roles? Is it present in other Atm or ABC transporters?

We included a new figure, “Figure 2 supplement 3”, comparing different homologs, *At*Atm3, human ABCB7 (an Alphafold model), human ABCB6, yeast Atm1 and prokaryotic Atm1. Among these five structures, only *At*Atm3 presents a loop between TM1 and TM2. A quick protein sequence blast in PubMed showed the presence of this loop in many other plant atm transporters. This long loop has not been observed in other structure we know so far, but an external helix was observed for PglK, a lipid-linked oligosaccharide flippase, and this loop has been implicated in substrate interaction (Perez et al., Nature 524, 433 (2015); doi: 10.1038/nature14953)

6) The cysteine-exclusion hypothesis of Atm3 is interesting, but lacks experimental validation. Please provide some additional experimental data to support this (e.g., is a mixed disulfide formed between GSH and cysteine(s) introduced into binding site based on locations in structural homologues) and/or appropriately tone down the statement to more closely reflect the presented data.

We agree this would be an important test of our hypothesis. We did introduce cysteine mutations in *Na*Atm1 at position M317, M320 and the double mutations in TM6 to test this model. Unfortunately, there was no expression of any of these mutants. We recognize that a negative result like this can have many explanations, and that testing other residues would be appropriate (including in *At*Atm3). At the time we were doing these studies, we were working under Covid restrictions and rather than testing additional residues, we focused on trapping additional conformational states.

Our apologies for the over-inflated claims about the mixed disulfides– we tried hard to appropriately describe our analysis, recognizing that this was based on inferences from sequence and structural comparisons (no cysteines in the glutathione binding site), but not from a direct test. We have reworded the abstract and page 13 to try to more accurately reflect the presented data.

Reviewer #1 (Recommendations for the authors):The cysteine-exclusion of Atm3 would benefit from some additional experimental analysis, such as the introduction of cysteine residues as found in the closest structural homologues harbouring cysteines and their impact on oxidised glutathione export.

Please see our response to essential revision 6.

Reviewer #2 (Recommendations for the authors):1) If this work is looked at in the larger context of type I ABC exporters (and in particular in the structure series of NaAtm1 published by the same authors in 2020 in PNAS), one burning question comes to my mind: How universal and generally relevant is what the authors call the closed conformation? In Fan et al., PNAS, 2020, the Rees lab for the first time (at least to the best of my knowledge) described a closed conformation for NaAtm1. The closed conformation has the NBDs fully dimerized/closed and the extracellular gate is closed, but in contrast to the more often observed outward-occluded conformation (first mentioning by Beis lab for McjD and ever since seen several times), there is no occluded cavity, but TM6-kinking results in a collapse of the internal cavity (thus called closed conformation). The major punch-line of the NaAtm1 story was that NaAtm1 assumes an outward-occluded conformation in the presence of non-hydrolysable AMP-PNP (pre-hydrolysis state), but a closed conformation in the presence of ATP-vanadate (post-hydrolysis state), suggesting a pumping/expelling mechanism that is dependent driven by ATP hydrolysis. Currently, the structural differentiation of AMP-PNP-bound versus ATP-vanadate bound state in light of the cavity collapse has only been described for NaAtm1 thus far, but not for any other type I ABC exporter. In line with the findings for NaAtm1, the authors describe in this paper the ATP-vanadate bound structure of AtAtm3 to assume again the closed conformation with kinked TMH6 and collapsed internal cavity (though, a small internal cavity still seems to exist according to Figure 3c). The obvious question (which does not appear to be mentioned in the paper) is whether the authors have attempted to obtain the AMP-PNP bound structure of AtAtm3 and if so, whether the outward-occluded conformation featuring an internal cavity was seen the way it was observed for NaAtm1?Overall, I feel there is still limited appreciation in the ABC transporter field for the interesting and unique closed conformation, likely owing to the fact that the discovery of the first closed conformation is comparatively recent. This paper is another chance to announce/emphasize the existence of the closed conformation. Ideally, this is addressed by further structures (though that's only a reasonable request if it is straight-forward to achieve). Alternative possibilities are biochemical experiments (e.g. cross-linking, or DEER).

Thank you – we have discussed this situation in our response to essential revision 1.

2) To stick to the "conformation business", I wonder whether the principal component analysis is superior to analyses simply measuring distances between residues at strategic positions (e.g. between coupling helices, at the extracellular gate and (to distinguish between closed and occluded) at the kinking prolines in TM6. Such distance measurements together with cavity determinations are in my view sufficient to distinguish between the relevant conformations and are more intuitive to grasp (see also PMID: 33098660).In particular the distance between the coupling helices informs about NBD separation/completeness of NBD closure, hence allowing to differentiate between inward-facing/inward-occluded states (or the special case of the unlock return state of TmrAB) and outward-facing, outward-occluded and (as relevant here) the closed conformation. The closed conformation seems to correspond to an outward-occluded conformation when it comes the degree of intracellular/extracellular gate closure, but does not feature an enclosed cavity. I would assume that an inter-TM6 distance measurement is able to distinguish outward-occluded from closed; this being the most important advantage over the principal component analysis, which fails to distinguish outward-facing/outward-occluded and closed (as evident from Figure 6a). And finally, the degree of extracellular gate opening is of general interest when analysing different conformations with fully closed NBDs.Unless I have overlooked something (which is well possible), I would consider such simple distance measurements as more informative than the principal component analysis.

Our motivation was to provide a single parameter that could be used to project transporter structures onto a single conformational axis. The principal component of the PCA provides this descriptor, but we agree that this is not a unique description of the protein conformation. We have explored the use of several alternative metrics based on distances between pairs of residues – some are highly (anti-) correlated with the PCA analysis, while others (like the distance between residues in the outer loops) better discriminate between outward facing conformations, but at the expense of inward-facing conformations. For the purposes of our qualitative analysis, that did not seem to be a critical distinction. Please see additional comments in our response to essential revision 2.

3) The authors state in the abstract that "The absence of structures containing both nucleotide and transport substrate suggests that this ternary complex exists only transiently during the transport cycle." This provokes the question whether cryo-EM analyses were carried out in the presence of ATP and GSSG, i.e. under hydrolysis conditions and in the presence of the substrate. It is well established in the field that outward-facing trapped ABC exporters (via vanadate trapping or EtoQ mutations at the Walker B motif) do not bind substrates (e.g. see MRP1 and ABCB1), hence it is not at all surprising that ATP-vanadate trapped AtAtm3 had no GSSG bound. However, the ternary complex may have been captured under hydrolysis conditions.

Please see our response to essential revision 3.

Reviewer #3 (Recommendations for the authors):1) In Figure 3, it could be helpful to label the functional states in panels in a, b, c, d, f, g, h. If the distances between 2 NBDs can be measured and labeled in panel a and b (apo and GSSG bound form), it would be more helpful for the non-ABC transporter audience to visualize the conformational change globally.

Thank you – please see our response to essential revision.

2) In Figure 6, the color codes in the legend do not match with those on the graph.

Thank you – please seen our response to essential revision.

3) In Figure S4, there are 2 high-resolution reconstructions of AtAtm3 in an inward-facing conformation with GSSG bound (3.57A and 3.68A respectively), are they different structurally?

Thank you – please seen our response to essential revision.